# Ultrasound-Guided Blocks for Spine Surgery: Part 1—Cervix

**DOI:** 10.3390/ijerph20032098

**Published:** 2023-01-23

**Authors:** Kamil Adamczyk, Kamil Koszela, Artur Zaczyński, Marcin Niedźwiecki, Sybilla Brzozowska-Mańkowska, Robert Gasik

**Affiliations:** 1Department of Anaesthesiology and Intensive Therapy, Central Clinical Hospital of the Ministry of Interior and Administration in Warsaw, 02-507 Warsaw, Poland; 2Department of Anaesthesiology, National Institute of Geriatrics, Rheumatology and Rehabilitation in Warsaw, 02-637 Warsaw, Poland; 3Neuroorthopedics and Neurology Clinic and Polyclinic, National Institute of Geriatrics, Rheumatology and Rehabilitation, 02-637 Warsaw, Poland; 4Department of Neurosurgery, Central Clinical Hospital of the Ministry of Interior and Administration in Warsaw, 02-507 Warsaw, Poland

**Keywords:** spine surgery, regional anesthesia, multimodal analgesia

## Abstract

Postoperative pain is common following spine surgery, particularly complex procedures. The main anesthetic efforts are focused on applying multimodal analgesia beforehand, and regional anesthesia is a critical component of it. The purpose of this study is to examine the existing techniques for regional anesthesia in cervical spine surgery and to determine their effect and safety on pain reduction and postoperative patient’s recovery. The electronic databases were searched for all literature pertaining to cervical nerve block procedures. The following peripheral, cervical nerve blocks were selected and described: paravertebral block, cervical plexus clock, paraspinal interfascial plane blocks such as multifidus cervicis, retrolaminar, inter-semispinal and interfacial, as well as erector spinae plane block and stellate ganglion block. Clinicians should choose more superficial techniques in the cervical region, as they have been shown to be comparably effective and less hazardous compared to paravertebral blocks

## 1. Introduction

Postoperative pain is common following spine surgery, particularly complex procedures, laminectomy at minimum of three levels, or scoliosis surgery [1], especially on the first day after surgery [2]. This severity of postoperative pain is directly related to the number of vertebrae that were operated on during the procedure [3]. Disc herniation and spinal foraminal stenosis are the most prevalent underlying pathologies of the cervical spine, occurring most often around C5–C7 levels [4,5,6,7]. While the stated mean duration of stay is roughly 2 days for ACDF (anterior cervical discectomy and fusion), several centers conduct this surgery as an outpatient service [8,9,10,11], as in disc arthroplasty [12]. In spine surgery, more adverse events, longer hospital stays, and higher expenses have been linked to increasing narcotic usage and insufficient pain management [13,14,15,16]. The main anesthetic efforts are focused on applying multimodal analgesia before, during and after spine surgery. This procedure is proven effective in reducing postoperative pain and faster recovery after surgery [13,14,17], and regional anesthesia is a critical component of it.

Surgical injury results in the production of many inflammatory mediators on the surgical site, which enter the circulation and disseminate systemically [18]. In addition to blocking the conduction of the pain impulse from the site of damage, local anesthetics also reduce the local inflammatory response [19,20]. However, research indicates that local anesthetic infiltration of a site following spine surgery is unsuccessful [21]. Therefore, regional anesthesia methods should be explored not just to increase patient comfort but also to decrease postoperative complications, such as digestion problems. Addressing these issues, which are the most prevalent causes of extended hospitalization and readmission following cervical spine procedures, can considerably decrease hospitalization duration and cost.

The purpose of this study is to examine the existing techniques for regional anesthesia in cervical spine surgery and to determine their effect and safety on pain reduction and postoperative patient’s recovery in a literature review manner.

## 2. Methodology

The search approach included exploring the electronic databases Medline, Pubmed, and Google Scholar for all literature pertaining to cervical nerve block procedures. The searched articles were further evaluated for papers of significance. The following keywords were used alone or in combination for databases search: “cervical spine surgery regional anesthesia”, “nerve block cervical region”, “peripheral nerve blocks cervical region”, “cervical paravertebral block”, “cervical plexus block”, “paravertebral brachial plexus block”, “cervical retrolaminar block”, “cervical erector spinae block”, “paraspinal interfascial plane block”, “multifidus cervicis plane block”, “retrolaminar cervical block”, “inter-semispinal plane block”, “cervical interfascial plane block”, and “stellate ganglion block”.

## 3. Cervical Paravertebral Block (CPVB)

Paravertebral block (PVB) is the injection of a local anesthetic into the paravertebral space—the area where the spinal nerves exit the intervertebral foramen. Unlike thoracic and lumbar nerve roots, cervical nerves emerge foramina above the vertebral bodies. The cervical paravertebral space is located at the posterior triangle of the neck formed by the deep cervical fascia, the paravertebral muscles, and the cervical vertebra [22].

The first description of the paravertebral block was published in 1905 by Hugo Sellheim, the head of Gynecology and Obstetrics at the University of Leipzig [23], while Kappis [24] gave the first description of a paravertebral block in the cervical region in 1923. In 1978, Eason and Wyatt [25] described a new approach for this blockage, which is still in use today: puncture 3 cm from the midline, contact the transverse process, and then proceed through its upper surface. The 2003 study by Boezaart et al. is the first modern report on the use of continuous cervical paravertebral block as a novel C6 paravertebral approach to the brachial plexus block [26]. In subsequent years, the use of ultrasound in cervical paravertebral blockades was again described as a modification of the brachial plexus block [27,28]. 

However, because of complications unique to this type of procedure, which include, first, intrathecal injection, as well as phrenic nerve paresis, this technique for analgesia for spine surgery is not yet extensively used and has been linked to mixed results. Currently, these blocks are used mostly for the diagnosis and management of chronic pain. Particularly, in individuals with cervical disc displacement, cervical foraminal stenosis, or cancer-related pain in the cervical spine or shoulder, cervical paravertebral nerve blocks may be beneficial for both diagnosis and treatment [29]. 

If local anesthetic injection is placed deep to the prevertebral fascia between C2 and C4 for the deep cervical plexus block, it is technically a paravertebral block, as in case of deep injection to the prevertebral fascia, between the 5th and 7th cervical vertebrae, for posterior approach to the interscalene brachial plexus block. 

The paravertebral block is a distinct subset of regional anesthetic techniques, as it differs significantly from both peripheral and central blockades. The method includes injecting a local anesthetic solution down the vertebral column, near to where the spinal nerves emerge, producing unilateral blockade of somatic and sympathetic nerves. It should also be noted that the dural sleeves follow the nerve roots into the paravertebral region [30]; consequently, dural puncture is a risk during paravertebral blocking. Clinicians should also be aware that the radicular arteries originating from the vertebral, ascending cervical, and deep cervical arteries are in close proximity to the spinal nerve in this location.

There are several ultrasonic blocking methods mentioned. Some writers injected local anesthetic between the prevertebral fascia and the cervical transverse process [31,32], whereas others injected after contacting the transverse process with the block needle [33,34]. Due to the anatomical differences between the paravertebral block and other peripheral blockades, as pointed out by Dr. Boezaart, a thin, flexible stimulating needle is unsuitable for a paravertebral block, while a large bore Tuohy needle for an epidural block should be used [30]. 

The most common complications after paravertebral nerve block in 620 adult patients were inadvertent vascular puncture, arterial hypotension, and epidural or intrathecal dispersion. It was calculated that the overall failure rate was 6.1% [35]. One of the most serious complications of this blockade described is the injection of a local anesthetic directly into the spinal cord with permanent loss of cervical spinal cord function [36]. There have been also reports of cardiac arrest following intravascular bupivacaine administration during cervical paravertebral blockade [37].

The authors were unable to identify a publication reporting the use of paravertebral block for postoperative pain management of the cervical spine surgery, except for the deep cervical plexus block, which is explained in further detail below. Elder et al. describe the perioperative placement of the catheter in the postoperative wound following posterior cervical spine fusion [38]. Continuous local anesthetic infusion decreased the use of opiates, patients reported less postoperative pain, were discharged home sooner, had a quicker first bowel movement, and were able to discontinue patient-controlled analgesia faster.

## 4. Cervical Plexus Block 

The cervical plexus arises from the C1–C4 cervical spinal nerves; however, the C1 nerve, the suboccipital nerve, is a motor neuron and is not targeted in any cervical plexus block technique [22]. The cervical plexus block (CPB) can be performed at three different depths: deep, intermediate, and superficial (Figure 1). The deep CPB is performed at the level of the C2–C4 spinous processes and, by definition, is a paravertebral block; the intermediate is performed between the deep cervical fascia and the prevertebral fascia, and the superficial is performed at the deep cervical fascia layer—targeting the lesser occipital, great auricular, transverse cervical, and supraclavicular nerves, four superficial branches of the cervical plexus.

The literature indicates that when a deep CPB block is performed at the single level, the adjacent levels are also anesthetized. In their case report, Boezaart et al., by injecting local anesthetic with contrast into the C4 paravertebral space, demonstrated that the medium extends from the base of the skull to the fifth cervical vertebra [39]. The technique of performing deep CPB with one or more injection points is well described in the literature, both having comparable efficacy, which is different only by the time required to achieve an effective blockade in case of a single administration [40].

A cervical plexus block has been linked to advantages such as reduced incidence of nausea and vomiting as well as a shorter surgical procedure and recovery time, according to a Chinese clinical trial comparing general anesthesia versus awake surgery with a deep cervical plexus block during anterior cervical discectomy and fusion (ACDF) [41]. Patients reported better comfort with general anesthetic than with regional anesthesia; however, shortly after surgery, cervical plexus anesthesia was associated with less pain than general anesthesia. 

Mariappan et al. evaluated the effect of superficial cervical plexus block (SCPB) on recovery following single- or two-level anterior cervical discectomy and fusion. Patients who had preoperative SCPB recovered better, although there were no changes in overall opioid consumption or discharge times. The SCPB was performed after the induction of general anesthesia, unilaterally with 10 mL of 0.25% bupivacaine, subcutaneously, in the middle of sternocleidomastoid muscle using fan-shaped technique [42]. 

## 5. Paraspinal Interfascial Plane Blocks (PIP)

In recent years, several new paraspinal blocks have been developed in which it has been demonstrated that the dorsal rami of cervical nerves can be blocked without the block needle entering the paravertebral space. After successfully introducing PIP blocks in thoracic and lumbar regions [43,44,45], novel cervical region blocks, including CIP (cervical interfascial plane) block—original article retracted [46] and MCP (multifidus cervicis plane) block [47], were reported in 2017. Since then, two additional cervical blocks—ISP (inter-semispinal plane) [48] in 2018 and retrolaminar [49] in 2021, have been described. All these interfascial plane blocks have shown promise as an alternative to neuraxial blockade for various procedures. Ultrasound-guided approaches to these blocks have been depictured in Figure 2 and shown in the cross-section in Figure 3. These blocks reduce the danger of spinal cord damage, epidural hematoma, and central infection. In contrast to paravertebral blocks, the injection site is far from the neuraxis, making it simpler and safer.

### 5.1. Multifidus Cervicis Plane Block (MCP)

First described by Ohgoshi et al. [47] as a case report of cervical laminoplasty perioperative analgesia, the MCP block was delivered bilaterally, under general anesthesia, by injecting 20 mL of 0.375% ropivacaine at each side, between the multifidus cervicis and semispinalis cervicis muscles fascial planes at C5 level. The patient’s recovery went well, and they needed no pain medication beyond the standard postoperative dose of loxoprofen sodium.

In another study, Mohamed et al. [50] evaluated the effectiveness of ultrasound-guided multifidus cervicis plane block with greater occipital nerve block in the treatment of cervicogenic headache, that is persistent hemicranial discomfort caused by a cervical spine condition that affects the C1, C2, and C3 cervical spinal nerves [51].

Sixty patients with cervicogenic headaches were recruited and divided into two block groups: greater occipital nerve block and ultrasound-guided multifidus cervicis plane block. Both blocks were effective treatments for cervicogenic headaches with a median VAS (Visual Analogue Scale) decrease of over 3 [50].

### 5.2. Retrolaminar Cervical Block

Retrolaminar blocks have mostly been studied for truncal surgery and pain disorders (thoracic and abdominal) [52,53]. Hochberg et al. performed cadaver research to examine the distribution of the contrast placed in the region posterior to the cervical lamina and conducted a clinical pilot study on patients with cervical radicular pain [54]. Each of the 12 patients was offered the option of undergoing cervical spine decompression surgery after failing conservative therapy for at least three months (i.e., rehabilitation and oral medications).

Patients were prone throughout procedure. Ultrasound and Doppler-guided needle insertion was performed, and fluoroscopy confirmed the spinal level. The needle was advanced using the in-plane method until it reached the lamina-spinous process junction. As a final stage, 4 mL of Lidocaine 0.5% was injected together with 10 mg (1mL) of dexamethasone. A cadaver investigation showed that injecting a contrast agent at the C6 level may reach cranial C2 and caudal T3 levels.

In another study, Khashan et al. combined cervical epidural steroid injections with retrolaminar blocks to treat radicular pain in the neck. The findings suggest that retrolaminar blocks could be used as a safer alternative to epidural steroid injections and decompressive surgery for radicular pain [55]. 

### 5.3. Inter-Semispinal Plane Block (ISP)

The inter-semispinal plane (ISP) block is an ultrasound-guided procedure that blocks the dorsal rami of the cervical spinal nerves by injecting local anesthetic into the fascial plane between the semispinalis cervicis and capitis muscles. According to the available evidence, this effectively alleviates pain following surgery. It was first described by Ohgoshi et al. [48] comparing the ISP block to the MCP block in eight healthy participants. In this study, both blockades showed similar effectiveness, area of action and duration of action. Researchers concluded that ISP could be an effective alternative to the MCP blockade.

In another study, Mostafa et al. [56] randomized participants to receive either general anesthesia alone (the control group) or bilateral ultrasound-guided ISP blocks at the C5 level with 20 mL of bupivacaine 0.25% in each side with general anesthesia (the ISP group) for posterior cervical spine surgery. After 30 min, 1 h, 2 h, 4 h, 6 h, 8 h, and 12 h postoperatively, the VAS scores of patients in the ISP group were considerably lower than those of patients in the control group.

## 6. Cervical Erector Spinae Plane Block

The semispinalis cervicis, the longissimus cervicis, and the iliocostalis cervicis muscles comprise the cervical section of the erector spinae muscle, and they all attach onto the C2–C6 TPs to bring the erector spinae plane all the way up to the cervical foramina at the base of the skull [57]. The block approach involves injecting an anesthetic into the deep fascia of the erector spine muscle, close to the transverse process (Figure 2 and Figure 3), and was first described by Forero et al. [58] in 2016 in the thoracic region. 

The erector spinae block was excluded from the PIP blocks in this study, as stated by Xu et al. [59], because it may target the ventral ramus and paravertebral region, whereas the PIP blocks only target the dorsal ramus of the spinal nerves.

In thoracic and abdominal surgical procedures, as well as in chronic neuropathic pain syndromes, the ESP (Erector Spinae Plane) block has been shown to produce good postoperative analgesia [60,61]. Nonetheless, anesthesiologists rarely perform this block in the cervical segment, with the thoracic segment being the most common location [61].

Although it is still a relatively uncommon technique, there have been reports demonstrating the safe and effective use of the blockade in patients undergoing cervical spine surgery [62,63].

Several techniques for performing the blockage have been described. Goyal et al. performed a bilateral ESP block under ultrasound guidance after administering general anesthesia and placing the patient in the prone position. The designated transverse process was located at or just below the incision for cervical surgery. Using the in-line technique, a 20G spinal needle was used to approach the transverse process and inject medication solution between the tip of the transverse process and the erector spinae muscle in a caudal-to-cranial trajectory [63].

It should be noted, however, that when performing an ESP block, there is a risk of nerve palsy. As a result, some authors argue that bilateral ESP blockage in the cervical region should be avoided [64]. Yuichi Ohgoshi, the creator of the aforementioned MCP and ISP blockades, also asserts [65] that in the event of cervical spine surgery, these blockades are preferable to ESP blockade due to the exclusive posterior spinal nerve block, as opposed to the potential of brachial plexus blockade in cervical ESP. Dr. Ohgoshi asserts that the semispinalis cervicis muscle is the anatomical boundary defining the local diffusion of a local anesthetic in MCP and ISP blockades. However, according to Dr. Tsui and Dr. Elsharkawy from Stanford University [66], it is not the semispinalis cervicis muscle that is the anatomical barrier for the local anesthetic but rather the patient’s individual anatomical predilection of the neck fascia. In addition, they contend that the anesthetic can penetrate the surrounding space when a higher volume is used, independent of the type of PIP block.

## 7. Stellate Ganglion Block

The stellate ganglion, located lateral to the trachea and esophagus, anterior to the longus colli muscle at C7, and posteromedial to the carotid sheath [68], is made up of the inferior cervical and first thoracic ganglia, and it provides sympathetic innervation to the upper extremities, head, neck, and heart. Angina pectoris, cardiac arrhythmias, bronchial asthma, primary hypertension, vascular headache, and chronic neuropathic pain may be diagnosed and treated with stellate ganglion block (SGB) [69,70,71,72,73].

SGB was first performed using a typical blind procedure [74,75] with injections of local anesthetic; however, this technique had a significant risk of adverse effects. SGB has been carried out using a variety of approaches, including anterior, lateral, and posterior, using several different techniques: fluoroscopy [76], CT [77,78] and recently ultrasounds [79,80,81]. Fluoroscopy became the standard of care for SGB; however, it has its own limitations, e.g., cannot see blood vessels near the stellate ganglion [82]. The block was more effective and safer with ultrasound guidance than with fluoroscopy guidance alone. Ultrasound, as opposed to fluoroscopy-guided blocking, prevents intravascular injections and soft tissue injury [80] (Figure 4).

Stellate ganglion block can be performed at either the C6 or C7 level, with C6 blockade resulting in more effective sympathetic blockade of the head and neck and less sympathetic blockade of the upper extremities, which may be desirable in some clinical situations [83].

The authors’ preferred access is lateral access. Doppler mode should be used to scan the block area for the presence of vascular structures: carotid artery, jugular vein and vertebral artery. Additionally, trachea, thyroid gland, longismus colli muscle and the transverse process of C6 or C7 should be visualized. Using the in-plane puncture method, the needle is advanced into the anterior scalene muscle, between the C6/C7 nerve root and the internal jugular vein, until it reaches the surface of the long cervical muscle [84].

It should be emphasized that the place of depositing the block is in close proximity to the root of the spinal nerve, so, as in the case of the paravertebral block, there is a risk of needle penetration of the dura mater. In addition, the vertebral artery runs in the blockage area. A case of cardiac arrest during stellate ganglion block with pencil-point nerve stimulator needle was described [85]. Therefore, a large bore Tuohy needle should be used for this block as with a paravertebral block.

The use of high doses of opioid drug in the postoperative period, especially after cervical spine surgery, contributes to bowel dysfunction [86]. In addition, surgical injuries and manipulations, which may be connected to imbalances between the sympathetic and parasympathetic systems, cause gastrointestinal disorders [87]. Probably via controlling the autonomic and immunological systems, blocking the stellate ganglion may improves digestive function after surgery [88,89].

Randomized controlled clinical trials on patients undergoing breast cancer surgery demonstrated improved sleep and digestive function during the postoperative period when stellate ganglion block was performed [90]. In another study, colon cancer patients following laparoscopic colorectal cancer surgery had their gastrointestinal functioning improved and stress levels reduced after SGB [91]. Finally, ultrasound-guided SGB increased patient satisfaction and expedited the restoration of digestion process following thoracolumbar spinal surgery [92].

Participation of the sympathetic nervous system in pain genesis has been best characterized in chronic and often neuropathic pain conditions, such as complicated regional pain syndrome; however, it is unclear whether SGB blockade has an effect on postoperative pain perception.

## 8. Conclusions

Due to the presence and technical advancement of ultrasound machines and fluoroscopy, CPVB-related side effects are occurring less frequently nowadays. Nevertheless, with the introduction of new techniques of interfascial plane blocks, clinicians should choose more superficial techniques in the cervical region, as they have been shown to be comparably effective and less hazardous compared to paravertebral blocks. As pointed out by Choquet et al., the deep block is more than twice as likely as the superficial block to generate a serious life-threatening problem as a result of block placement [93]. 

The current literature, while limited, indicates that three paraspinal plane blocks, such as multifidus cervicis plane block, retrolaminar cervical block or inter-semispinal plane block, and besides that erector spinae plane block, have a high safety profile and effectiveness, but the latter appears to have an increased risk of adverse effects such as phrenic nerve palsy.

Stellate ganglion block may be an interesting addition to postoperative care following cervical spine surgeries to reduce the negative effects of postoperative pharmacotherapy, particularly on the digestive system. However, because of the close proximity of vascular structures, particularly the vertebral artery and spinal nerve roots, special precautions, akin to a paravertebral block, should be taken.

The ease of performance, high efficiency of the block, and minimal risk of side effects, especially with paraspinal blocks, should encourage anesthesiologists to implement these techniques of regional anesthesia in cervical spine surgery in everyday practice.

## Figures and Tables

**Figure 1 ijerph-20-02098-f001:**
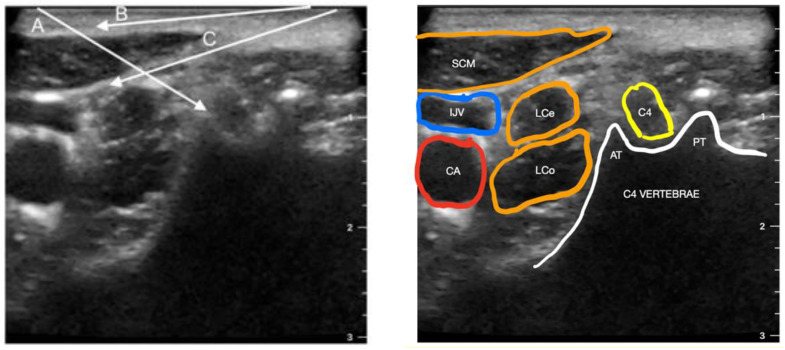
Transverse sonogram of the lateral cervical area at the C4 level. The arrows indicate the needle trajectories for cervical plexus block: A—deep; B—superficial; C—intermediate. SCM—sternocleidomastoid muscle; IJV—internal jugular vein; CA—carotid artery; LCe—longus cervicis muscle; LCa—longus colli muscle; AT—anterior tubercle of transverse process; PT—posterior tubercle of transverse process.

**Figure 2 ijerph-20-02098-f002:**
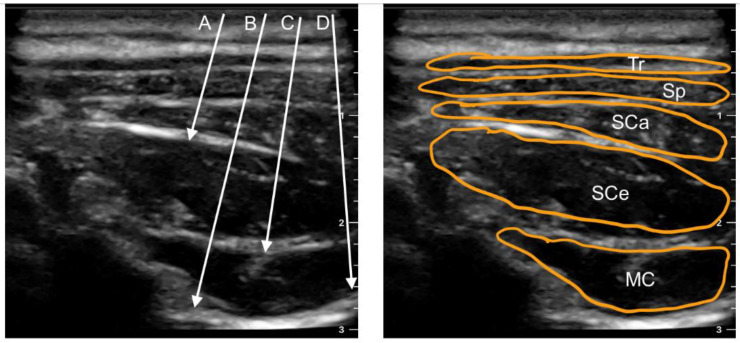
Transverse sonogram of the posterior neck area at the C5 level. The arrows indicate the needle trajectories for: A—inter-semispinal plane block; B—retrolaminar cervical block; C—multifidus cervicis plane block; D—cervical erector spinae plane block. MC—multifidus cervicis muscle; SCe—semispinalis cervicis muscle; SCa—semispinalis capitis muscle; Sp—splenius muscle; Tr—trapezius muscle.

**Figure 3 ijerph-20-02098-f003:**
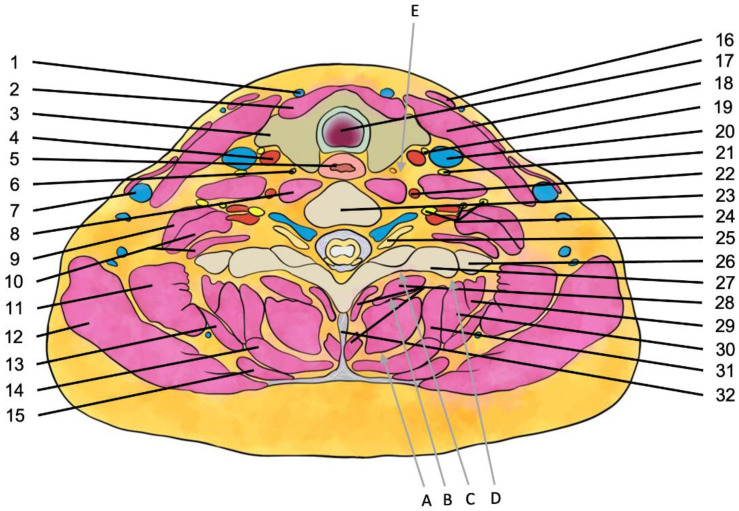
Cross-section of the neck at the level of C7/C8. The arrows indicate the needle trajectories for: A—inter-semispinal plane block; B—multifidus cervicis plane block; C—retrolaminar cervical block; D—cervical erector spinae plane block; E—medial transthyroid stellate ganglion block. 1—anterior jugular vein; 2—sternothyoid muscle; 3—thyroid gland; 4—common carotid artery; 5—sympathetic trunk; 6—sympathetic trunk; 7—external jugular vein; 8—longus colli muscle; 9—middle scalene muscle; 10—posterior scalene muscle; 11—levator scapulae muscle; 12—trapezius muscle; 13—serratus posterior superior muscle; 14—splenius capitis muscle; 15—rhomboid minor muscle; 16—platysma; 17—sternocleidomastoid muscle; 18—trachea; 19—vagus nerve; 20—internal jugular vein; 21—phrenic nerve; 22—vertebral artery; 23—cervical vertebra (C7); 24—Spinal nerves (C5, C6, and C7); 25—Spinal nerve root (C8); 26—First rib; 27—transverse process of vertebra; 28—multifidus muscle; 29—iliocostalis cervicis muscle; 30—longissimus cervicis muscle; 31—splenius cervicis muscle; 32—interspinous ligament. Scheme based on [67].

**Figure 4 ijerph-20-02098-f004:**
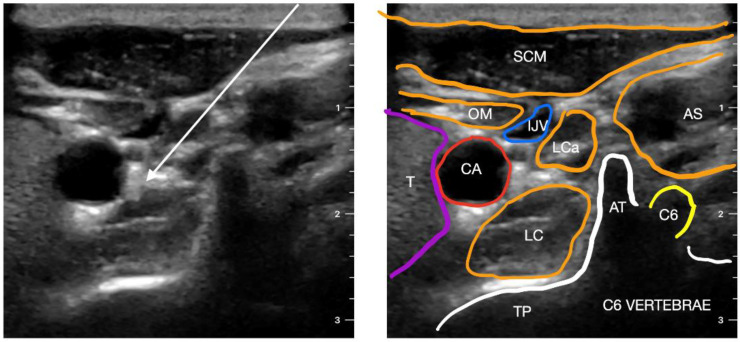
Transverse sonogram of the posterior neck area at the C6 level. The arrow indicates the needle trajectories for stellate ganglion block. SCM—sternocleidomastoid muscle; OM—omohyoid muscle; T—thyroid gland; CA—carotid artery; IJV— internal jugular vein; LC—longus colli muscle; LCa—longus capitis muscle; AS—anterior scalene muscle; TP—transverse process; AT—anterior tubercle of the transverse process; C6—C6 nerve root.

## Data Availability

Not applicable.

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
