# Peer review of "Ultrasound-Guided Blocks for Spine Surgery: Part 1—Cervix"

_ijerph, 2023, doi:10.3390/ijerph20032098_

Round 1
Reviewer 1 Report
Dear authors: Thank you for this interesting review manuscript. There are a few typos/incorrect figure references. In Figure 1, the longus cervicis muscle ought to be labeled as LCe however in the figure it is labeled "LCa". Section 4.2 "Retrolaminar Cervical block there is an improper citation "Hochberg and al. performed cadaver research to exam...". This ought to read 'et al', not 'and al'.
Line 195 "Each of the 9 patient"...this should be pluralized.
4.4. Cervical interfascial plane (CIP) block-this technique and report is cited as containing fabricated/falsified data and ought not even be presented.
Otherwise the paper is well-written and nicely stipulates that due to the published literature on various techniques used to garner cervical spine regional anaesthesia and to minimize post op complications that the more superficial routes of anaesthesia ought to promoted. This will help anesthetists/pain specialists in their choice of regional anaesthesia as an adjunct to general anaesthesia (when chosen).
Author Response
Dear Reviewer,
thank you very much for all your valuable comment. We have corrected the article as you suggest.
In Figure 1, the longus cervicis muscle ought to be labeled as LCe however in the figure it is labeled "LCa".
It was corrected.
Section 4.2 "Retrolaminar Cervical block there is an improper citation "Hochberg and al. performed cadaver research to exam...". This ought to read 'et al', not 'and al'.
It was corrected.
Line 195 "Each of the 9 patient"...this should be pluralized.
It was corrected.
4.4. Cervical interfascial plane (CIP) block-this technique and report is cited as containing fabricated/falsified data and ought not even be presented.
It has been removed.
Reviewer 2 Report
Manuscript ID: ijerph-2121910-peer-review-v1
Manuscript title: Ultrasound-guided blocks for spine surgery: part 1 – cervix
Comments
This manuscript reports a study designed to examine the existing techniques for regional anesthesia in cervical spine surgery and to determine their effect and safety on pain reduction and postoperative patient's recovery. The manuscript is well-written and organized in a scientific style for a review. The content seems relevant to the field. I have only a few comments for the authors to consider.
Major comments
1. Introduction, lines 54-63. This paragraph briefly describes the procedures for (systematic) literature review, although the type of review is not reported in this text. Even for a narrative literature review, please consider reporting this text in a separate section to make explicit for readers the type of review. Complementarily, this choice of method must be discussed later on the text (please see [https://doi.org/10.1111/j.1471-1842.2009.00848.x] for some strengths and weaknesses of several types of reviews).
Minor comments
1. Introduction, line 30. Possible typo (This verity -> The severity?).
Author Response
Dear Reviewer,
thank you very much for all your valuable comment. We have corrected the article as you suggest.
Introduction, lines 54-63. This paragraph briefly describes the procedures for (systematic) literature review, although the type of review is not reported in this text. Even for a narrative literature review, please consider reporting this text in a separate section to make explicit for readers the type of review. Complementarily, this choice of method must be discussed later on the text (please see [https://doi.org/10.1111/j.1471-1842.2009.00848.x] for some strengths and weaknesses of several types of reviews).
It was corrected.
1. Introduction, line 30. Possible typo (This verity -> The severity?).
It was corrected.
Round 2
Reviewer 2 Report
Thank you for submitting a revised version of your manuscript. All my comments were addressed. I have no new comments.